# Predictive machine learning models for ascending aortic dilatation in patients with bicuspid and tricuspid aortic valves undergoing cardiothoracic surgery: a prospective, single-centre and observational study

Bamba Gaye [1], Maxime Vignac,[1] Jesper R Gådin,[1] Magalie Ladouceur,[2] Kenneth Caidahl,[3] Christian Olsson [4], Anders Franco-Cereceda,[4] Per Eriksson,[1] Hanna M Björck [1]

For numbered affiliations see end of article.

**Correspondence to**
Dr Hanna M Björck;
Hanna.Bjorck@ki.se and
Dr Bamba Gaye;
bamba.gaye@inserm.fr

## ABSTRACT

**Objectives** The objective of this study was to develop clinical classifiers aiming to identify prevalent ascending aortic dilatation in patients with bicuspid aortic valve (BAV) and tricuspid aortic valve (TAV).

**Design and setting** A prospective, single-centre and observational cohort.

**Participants** The study involved 543 BAV and 491 TAV patients with aortic valve disease and/or ascending aortic dilatation, excluding those with coronary artery disease, undergoing cardiothoracic surgery at the Karolinska University Hospital (Sweden).

**Main outcome measures** Predictors of high risk of ascending aortic dilatation (defined as ascending aorta with a diameter above 40 mm) were identified through the application of machine learning algorithms and classic logistic regression models.

**Exposures** Comprehensive multidimensional data, including valve morphology, clinical information, family history of cardiovascular diseases, prevalent diseases, demographic details, lifestyle factors, and medication.

**Results** BAV patients, with an average age of 60.4±12.4 years, showed a higher frequency of aortic dilatation (45.3%) compared with TAV patients, who had an average age of 70.4±9.1 years (28.9% dilatation, p <0.001). Aneurysm prediction models for TAV patients exhibited mean area under the receiver-operating-characteristic curve (AUC) values above 0.8, with the absence of aortic stenosis being the primary predictor, followed by diabetes and high-sensitivity C reactive protein. Conversely, prediction models for BAV patients resulted in AUC values between 0.5 and 0.55, indicating low usefulness for predicting aortic dilatation. Classification results remained consistent across all machine learning algorithms and classic logistic regression models.

**Conclusion and recommendation** Cardiovascular risk profiles appear to be more predictive of aortopathy in TAV patients than in patients with BAV. This adds evidence to the fact that BAV-associated and TAV-associated aortopathy involves different pathways to aneurysm

## STRENGTHS AND LIMITATIONS OF THIS STUDY

⇒ Comprehensive clinical and epidemiological data from 1034 individuals (543 bicuspid aortic valve (BAV) and 491 tricuspid aortic valve (TAV)) were analysed, providing a robust dataset for examination.

⇒ Aortic valve morphology assessment during open-heart surgery adds a significant strength, enhancing reliability compared with relying solely on echocardiography.

⇒ However, the study has inherent limitations, including potential selection bias due to the exclusion of individuals with significant coronary artery disease, potentially leading to an overestimation of aneurysm prevalence.

⇒ Being a surgical cohort, the study population may include individuals with worse outcomes, which could affect generalisability to counterparts of similar age in non-surgical settings.

⇒ The study is limited to a single centre and mono-ethnicity, potentially impacting the generalisability of the findings to broader BAV/TAV populations.

formation and highlights the need for specific aneurysm preventions in these patients. Further, our results highlight that machine learning approaches do not outperform classical prediction methods in addressing complex interactions and non-linear relations between variables.

## INTRODUCTION

Thoracic aortic aneurysm (TAA) is a silent disease characterised by medial degradation and pathological widening of the intrathoracic aorta. Individuals with a bicuspid aortic valve (BAV) are at higher risk of developing ascending aortic aneurysm than individuals born with a normal tricuspid aortic valve (TAV).[1 2] BAV is the most common congenital

cardiac abnormality with an incidence of about 13 per 1000 live births and more than one-third of all BAV patients develop aneurysm or aortic dissection later in life.[3–5] The underlying mechanism is not known although a turbulent flow, caused by the abnormal bicuspid valve anatomy, and/or a genetic defect, has been proposed as contributing factors. The complications of BAV have led to a number of unanswered clinical questions such as: (1) what mechanisms are underlying the development of aortopathy in adults with BAV, (2) how to predict the development of aortopathy in individuals with BAV and (3) will prediction models performance and predictors be different in patients with BAV compared with patients with TAV. Echocardiography is effective in showing the degree of the sinus valsava and the annulus dilatation. However, it is more difficult to know the degree of dilation of the sinotubular junction and the ascending aorta even though they are the more frequently dilatated sections in patients with BAV.[6]

Our aim was to identify clinical classifiers of prevalent aortopathy using comprehensive multidimensional clinical data. Automated machine learning methods and traditional regression models were performed on a total of 1034 subjects with either BAV or TAV.

## METHODS
### Clinical cohorts
The Advanced Study of Aortic Pathology (ASAP) study design and description have been previously published.[1 7] Briefly, the ASAP cohort is a single-centre, observational cohort study of consecutive patients with aortic valve and/or ascending aortic disease undergoing elective open-heart surgery at the Cardiothoracic Surgery Unit, Karolinska University Hospital in Stockholm, Sweden. Inclusion criteria were patients aged 18 or above with aortic valve disease (ie, aortic stenosis (AS) or aortic regurgitation (AR)) and/or ascending aorta dilatation (aneurysm or ectasia) but devoid of significant coronary artery disease (stenosis>70% or fractional flow reserve <0.80) and primarily not planned for another concomitant valve surgery. The Disease of the Aortic Valve Ascending Aorta and Coronary Arteries (DAVAACA) study was set up in the continuity of the ASAP study. The ASAP/DAVAACA study started in February 2007 and includes multidimensional data (blood analyses, genetic and clinical data, family history of cardiovascular diseases (CVDs), prevalent diseases data, demographic characteristics, lifestyle habits data and medication). In the present study, 1180 operated patients were included (28 June 2017). Exclusion of patients with unicuspid aortic valves, patients with missing data for cuspidity and aortic dilatation and patients with syndromic forms of TAA resulted in a final study population of 1034 subjects (online supplemental efigure 1). Data collection and classification of AS and AR have been previously described.[1] Aortic valve function and aortic dimensions were assessed according to the

standards outlined by the American Society of Echocardiography (ASE).[8–10]

### Echocardiography
Preoperative transthoracic echocardiography was used to determine aortic valve function. Transoesophageal echocardiography (TEE) was performed on the operating table, prior to surgery, under general anaesthesia and as previously described.[11] The echocardiographic evaluation has been already published elsewhere.[1] TEE evaluation was used to assess valvular function, aortic root morphology and ascending aortic diameter; measurements were performed according to standards outlined by the ASE. Diameter of the ascending aorta was measured at several locations (annulus, sinus Valsalva, STJ). An ascending aorta with a diameter exceeding 40 mm was categorised as aortic dilatation in both women and men. Similarly, patients with root dilatation exceeding 40 mm were classified as having aortic dilatation. Body surface area (BSA) was calculated using the Du Bois method.[12] AS and AR were defined according to standard guidelines for valve surgery.[13]

### Aortic valve morphology
The morphology of the aortic valve was determined by inspection of the valve during surgery. Based on appearance, the valve was classified according to number of cusps and commissures. Three cusps and three commissures denote a TAV; two cusps and two commissures denote a BAV (if a remnant commissural raphe was present) or true BAV (if no raphe was present). The BAV was further classified according to cusps fusion; right and left coronary cusps (RL), right and non-coronary cusps (RN) or left and non-coronary cusps (LN).[1] The Sievers classification,[14] based on intraoperative analysis of the aortic valve morphology, was used to diagnose and type the BAV.

### Statistical analysis
Characteristics of the population were described using analysis of variance or $\chi^2$ tests when appropriate. Several meaningful blocks of variables were chosen (clinical data, family history of CVDs, prevalent diseases, demographic characteristics, lifestyle habits data and medication). Principal component analysis (PCA) and multiple correspondence analysis were used to explain the variance–covariance structure of the variables combined. To identify variables that separated dilated versus nondilated BAV and TAV patients, logistic regression models were used and automated machine learning algorithms were applied.

### Logistic regression
For logistic regression, a set of variables were selected to be included in the model. Since model performance can rely heavily on variable selection, different variable selection methods were tested prior to the logistic regression analysis (online supplemental efigure 2). First, variables were selected based on prior knowledge and/or biological plausibility. Then, two automated variable selection

methods were considered: (1) backward and forward elimination to optimise Akaike Information Criteria,[11] and (2) and least absolute shrinkage and selection operator (LASSO).[12]

## Machine learning algorithms

We used two machine learning algorithms, Random Forests and Artificial Neural Network, which are among the most widely and successfully used for clinical data.[15 16] Each one of them represents a different algorithm 'family', with different internal algorithm structures.[17] Since it was not known beforehand which kind of algorithm would perform best, algorithms with different internal structures were chosen to increase the probability of good discriminative performance. A 10-fold cross-validation method to our logistic regression was also applied, allowing us to split the training data into five subsets. Then, five times iterations over the five subsets were performed, such that the subset with the index same as the iteration number was used as the validation test and the remaining four subsets were training tests. Sensitivity, specificity and positive and negative predictive values for each predictive model were also assessed, applying a leave-one-out cross-validation method (online supplemental efigures 3 and 4).

## Patient and public involvement

It was not possible to involve patients or the public in the design, or conduct, or reporting, or dissemination plans of our research.

## RESULTS
## Characteristics of BAV and TAV patients

The study population included a total of 1034 patients (52% BAV and 48% TAV) (online supplemental etable 1 and efigure 1). BAV patients were significantly younger (60.4 (SD 12.40) years versus 70.4 (SD 9.09) years in TAV patients), and mainly men (73.1%) compared with TAV patients (63.3%). Furthermore, BAV patients had significantly higher BSA but, in contrast, a lower body mass index (BMI). AS was more common in BAV compared with TAV whereas no significant difference in the proportion of AR was found, even if the prevalence of AR was slightly higher in TAV patients. Finally, a higher prevalence of CVD history was observed in TAV individuals (hypertension, coronary artery disease, heart failure and stroke) and a higher percentage of family history of CVD (sibling history of myocardial infarction before 65 years of age).

## Aortic dilatation in BAV and TAV patients

Globally, PCA showed no separation among BAV patients with (BAV-D) or without (BAV-ND) ascending aortic dilatation (figure 1A), using general clinical characteristics, family history of CVD, prevalent diseases and demographics as input. By contrast, TAV patients with a dilated (TAV-D) ascending aorta clearly separated from TAV individuals with non-dilated (TAV-ND) ascending aortas

(figure 1B), indicating a more pronounced difference between these groups.

Analysis of patient descriptive data showed a pattern of significantly different associated traits between BAV-ND versus BAV-D and TAV-ND versus TAV-D (table 1, online supplemental etable 2) patients. Specifically, TAV-D patients were younger than TAV-ND patients while in the BAV group age did not differ. BAV-D patients, on the other hand, had a higher BSA and diastolic blood pressure, and a lower pulse pressure (PP) than BAV-ND patients. In both BAV and TAV patients, AR was more commonly associated with aortic dilatation, while AS was more prevalent in patients with non-dilated aorta. It was also noted that aortic dilatation was more localised to the tubular part of the ascending aorta in BAV patients whereas in TAV patients the dilatation more often included the aortic root (table 1). A positive association between dilatation and abdominal aortic aneurysm (AAA), and negative associations between dilatation and diabetes, and dilatation and angina were found in both BAV and TAV patients (table 1).

To further analyse BAV-associated and TAV-associated aortopathy, a multivariable logistic regression model was performed, including age, sex, BSA, low-density lipoprotein (LDL) cholesterol and high-sensitivity C reactive protein as covariates (table 2). Independent variables were chosen based on previous analyses with significant differences within groups. Covariates were selected based on stepwise selection and decision trees. The results showed that in BAV patients, AS, PP and diabetes were negatively associated with dilatation (adjusted models for age, sex, BSA, LDL and hsCRP were with AS: OR=0.44 (0.28 to 0.70), p=0.001; PP: OR=0.99 (0.98 to 1.0), p=0.025, and diabetes: OR=0.32 (0.16 to 0.62), p=0.001). Similarly, in TAV patients, dilatation was negatively associated with AS: OR=0.03 (0.02 to 0.06), p<0.001 and diabetes OR=0.11 (0.03 to 0.32), p<0.001. The magnitude of the association with AS was however 15-fold higher in TAV than in BAV patients. AR was positively associated with aortic dilation in both BAV and TAV with a fivefold higher magnitude in TAV compared with BAV patients (table 2). The ROC curve for the risk prediction model across the predictive methods without AS as predictor is shown in online supplemental efigure 5.

## Clinical classifiers of aortic dilatation

In order to answer the question if the prevalence of aortopathy in TAV and BAV patients may be identified using clinical characteristics, automated machine learning algorithms, including random forests and artificial neural network, and classic logistic regression models were used. This showed that the discrimination between TAV-ND and TAV-D patients was good for all models tested, with mean AUCs ranging from 0.81 to 0.88 (figure 2, online supplemental etable 3). The best discrimination was obtained using Random Forest (mean AUC: 0.88 (95% CI 0.78 to 0.96)) and GLM (mean AUC: 0.81 (95% CI 0.68 to 0.93)) (figure 2, online supplemental etable 3).

## A BAV patients

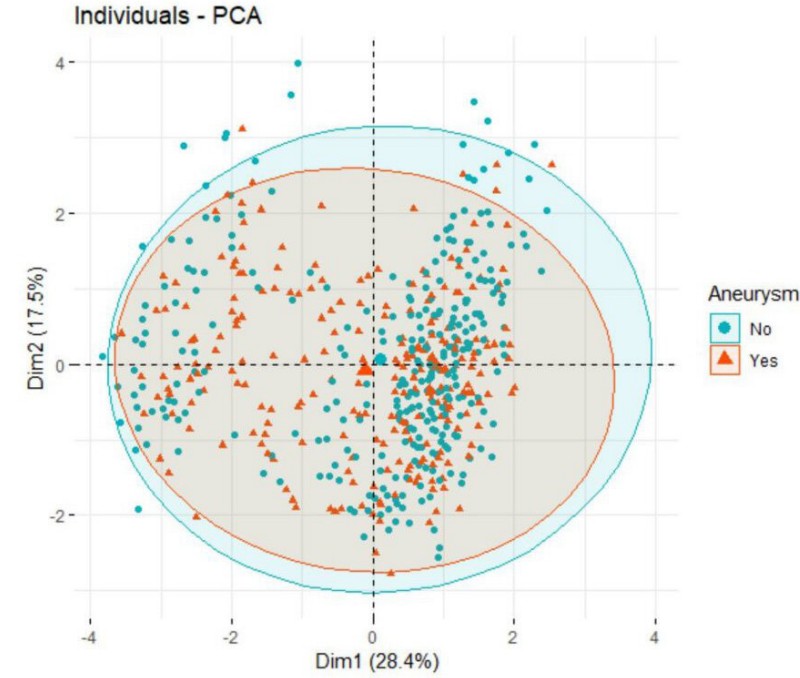

## B TAV patients

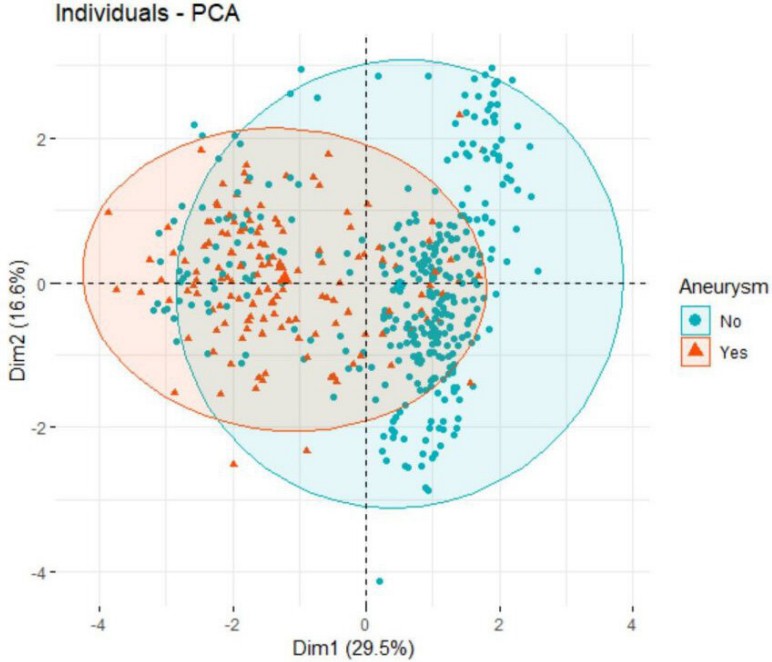

**Figure 1** Principal component analysis (PCA) plot of bicuspid aortic valve (BAV) versus tricuspid aortic valve (TAV) patients (with or without aneurysm).

Generalised linear regressions included AS, AR, PP, prevalence of diabetes, age, sex, BSA, LDL, hsCRP and BMI as variables. It was not possible to predict aortic dilatation in BAV individuals that are at high risk of developing aneurysm, as indicated by poor discrimination by all classifications models used (figure 2, online supplemental etable 3). This is in line with the lack of separation in the PCA.

Of note, the robustness of our classification models is supported by the results of the sensitivity analyses using imputed data (online supplemental etables 4 and 5). Online supplemental etable 5 indicates that the main characteristics between subjects with and without missing data did not differ substantially, minimising selection bias.

## DISCUSSION

In our analysis of 1034 Swedish subjects with BAVs and TAVs with and without prevalent aortopathy but devoid

**Table 1** Baseline characteristics of patients with bicuspid aortic valve (BAV) or tricuspid aortic valve (TAV) (dilated vs non-dilated ascending aorta)

| | BAV patients (n=543) | | | TAV patients (n=491) | | |
|---|---|---|---|---|---|---|
| | Dilated (n=246) | Non-dilated (n=297) | P value | Dilated (n=142) | Non-dilated (n=349) | P value |
| Gender: male | 182 (74.0%) | 215 (72.4%) | 0.749 | 89 (62.7%) | 222 (63.6%) | 0.927 |
| Age | 61.1 (11.2) | 59.8 (13.3) | 0.204 | 67.2 (10.1) | 71.8 (8.28) | <0.001 |
| Height | 177 (9.14) | 174 (9.29) | 0.002 | 174 (9.76) | 172 (9.77) | 0.043 |
| Weight | 83.1 (15.6) | 80.9 (15.8) | 0.100 | 81.4 (16.3) | 81.6 (15.7) | 0.909 |
| BSA | 2.00 (0.21) | 1.96 (0.22) | 0.052 | 1.95 (0.22) | 1.94 (0.22) | 0.479 |
| BMI | 26.5 (4.15) | 26.6 (4.36) | 0.916 | 26.9 (4.41) | 27.7 (4.90) | 0.087 |
| Regular smoker: | | | 0.176 | | | 0.537 |
| No | 108 (44.1%) | 142 (48.8%) | | 61 (43.3%) | 152 (43.8%) | |
| Former | 117 (47.8%) | 117 (40.2%) | | 69 (48.9%) | 177 (51.0%) | |
| Yes | 20 (8.16%) | 32 (11.0%) | | 11 (7.80%) | 18 (5.19%) | |
| Raphe: | | | 0.568 | | | |
| True BAV | 24 (9.96%) | 23 (7.96%) | | | | |
| Left-non-coronary | 1 (0.41%) | 4 (1.38%) | | | | |
| Right-non-coronary | 43 (17.8%) | 47 (16.3%) | | | | |
| Right–left | 173 (71.8%) | 215 (74.4%) | | | | |
| Systolic blood pressure | 135 (18.2) | 136 (18.9) | 0.299 | 144 (17.4) | 142 (21.5) | 0.299 |
| Diastolic blood pressure | 81.3 (12.3) | 79.0 (11.6) | 0.026 | 75.1 (13.0) | 76.5 (13.3) | 0.290 |
| Pulse pressure (PP) | 53.3 (16.5) | 57.2 (18.3) | 0.009 | 68.7 (18.1) | 65.4 (20.3) | 0.076 |
| Leukocytes | 5.70(4.70; 6.80) | 5.90(4.97; 7.00) | 0.052 | 6.00(5.00; 7.30) | 6.10(5.10; 7.40) | 0.386 |
| hsCRP | 0.93(0.50; 2.15) | 1.20(0.59; 2.80) | 0.056 | 2.10(0.98; 4.05) | 1.60(0.71; 3.80) | 0.050 |
| Cholesterol | 4.90(4.20; 5.70) | 4.80(4.20; 5.60) | 0.566 | 4.80(4.12; 5.40) | 4.60(3.90; 5.60) | 0.175 |
| LDL | 2.90(2.20; 3.60) | 2.80(2.30; 3.60) | 0.813 | 2.90(2.20; 3.50) | 2.60(1.90; 3.40) | 0.064 |
| Aortic stenosis: | 169 (68.7%) | 235 (79.1%) | 0.008 | 23 (16.2%) | 283 (81.1%) | <0.001 |
| Aortic regurgitation: | 83 (33.7%) | 75 (25.3%) | 0.038 | 93 (65.5%) | 67 (19.2%) | <0.001 |
| Diameter aortic anulus | 25.5 (3.41) | 24.2 (3.58) | <0.001 | 23.5 (3.17) | 22.8 (3.03) | 0.042 |
| Diameter sinus valsalva | 38.1 (5.38) | 33.6 (5.11) | <0.001 | 40.1 (6.98) | 32.8 (5.71) | <0.001 |
| Diameter STJ | 33.6 (5.63) | 28.2 (4.51) | <0.001 | 35.9 (6.76) | 26.6 (4.36) | <0.001 |
| Diameter ascend aorta | 46.9 (5.02) | 33.0 (4.27) | <0.001 | 50.5 (7.90) | 31.6 (4.03) | <0.001 |
| Sibling MI before 65 years | 13 (5.56%) | 19 (6.81%) | 0.688 | 11 (8.09%) | 35 (10.8%) | 0.474 |
| Mother MI before 65 years | 15 (6.20%) | 10 (3.64%) | 0.250 | 7 (5.26%) | 25 (7.67%) | 0.474 |
| Father MI before 65 years | 24 (10.6%) | 45 (16.3%) | 0.087 | 18 (14.1%) | 45 (14.4%) | 0.922 |
| Myocardial infarction | 8 (3.25%) | 11 (3.77%) | 0.930 | 9 (6.43%) | 33 (9.46%) | 0.367 |
| Stroke | 15 (6.10%) | 15 (5.08%) | 0.746 | 14 (10.0%) | 43 (12.3%) | 0.571 |
| AAA | 23 (9.39%) | 2 (0.68%) | <0.001 | 25 (18.2%) | 5 (1.44%) | <0.001 |
| Angina pectoris | 9 (3.66%) | 26 (8.87%) | 0.023 | 9 (6.43%) | 52 (14.9%) | 0.015 |
| Heart failure | 15 (6.15%) | 25 (8.65%) | 0.354 | 15 (10.6%) | 42 (12.2%) | 0.747 |
| Hypertension | 115 (47.1%) | 140 (47.6%) | 0.979 | 88 (62.9%) | 223 (65.0%) | 0.730 |
| Diabetes | 15 (6.10%) | 39 (13.3%) | 0.009 | 3 (2.16%) | 61 (17.5%) | <0.001 |
| Antihypertensive drugs | | | | | | |
| ACE inhibitor | 46 (18.7%) | 57 (19.3%) | 0.941 | 45 (31.9%) | 97 (27.9%) | 0.434 |
| Aspirin | 48 (19.5%) | 84 (28.5%) | 0.021 | 38 (27.0%) | 155 (44.8%) | 0.001 |
| Betablocker | 92 (37.4%) | 125 (42.4%) | 0.277 | 76 (53.9%) | 179 (51.3%) | 0.672 |

$\chi^2$ test was used for categorical variables; analysis of variance for normal-distributed continuous variables; Kruskal-Wallis test for non-normal continuous variables.
AAA, abdominal aortic aneurysm; BMI, body mass index; BSA, body surface area; hsCRP, high-sensitive C reactive protein; LDL, low-density lipoprotein; STJ, sinotubular junction.

**Table 2** Predictors of ascending aortic dilatation in bicuspid aortic valve (BAV) and tricuspid aortic valve (TAV) patients, separately

| | BAV | | | | TAV | | | | | |
|---|---|---|---|---|---|---|---|---|---|---|
| | Unadjusted model | Model 1 | Model 2 | | Unadjusted model | | Model 1 | | Model 2 | |
| | OR (95% CI)* | P value | OR (95% CI)* | P value | OR (95% CI)* | P value | OR (95% CI)* | P value | OR (95% CI)* | P value |
| Aortic stenosis | 0.58 (0.39 to 0.85) | 0.006 | 0.46 (0.24 to 0.88) | 0.001 | 0.05 (0.03 to 0.07) | 0.020 | 0.03 (0.02 to 0.06) | <0.001 | 0.02 (0.01 to 0.05) | <0.001 |
| Aortic regurgitation | 1.51 (1.04 to 2.19) | 0.031 | 1.42 (0.76 to 2.66) | 0.003 | 7.99 (5.19 to 12.45) | 0.270 | 7.47 (4.62 to 12.27) | <0.001 | 0.58 (0.23 to 1.36) | 0.229 |
| Pulse pressure | 0.99 (0.98 to 1.00) | 0.011 | 0.98 (0.97 to 0.99) | 0.025 | 1.01 (1.00 to 1.02) | 0.001 | 1.00 (0.99 to 1.02) | 0.091 | 0.99 (0.97 to 1.00) | 0.096 |
| Diabetes | 0.42 (0.22 to 0.77) | 0.007 | 0.36 (0.18 to 0.72) | 0.001 | 0.1 (0.03 to 0.29) | 0.005 | 0.11 (0.03 to 0.32) | <0.001 | 0.26 (0.06 to 0.87) | 0.048 |

Model 1: adjusted for Sex_Male+Age + BSA + LDL + hsCRP.
Model 2: adjusted for Sex_Male+Age + BSA + LDL+ hsCRP + the three other variables (ie, aortic regurgitation, pulse pressure and diabetes for aortic stenosis).
Univariate and multivariate logistic regression analysis.
*OR and 95% CI limits were obtained by logistic regression.

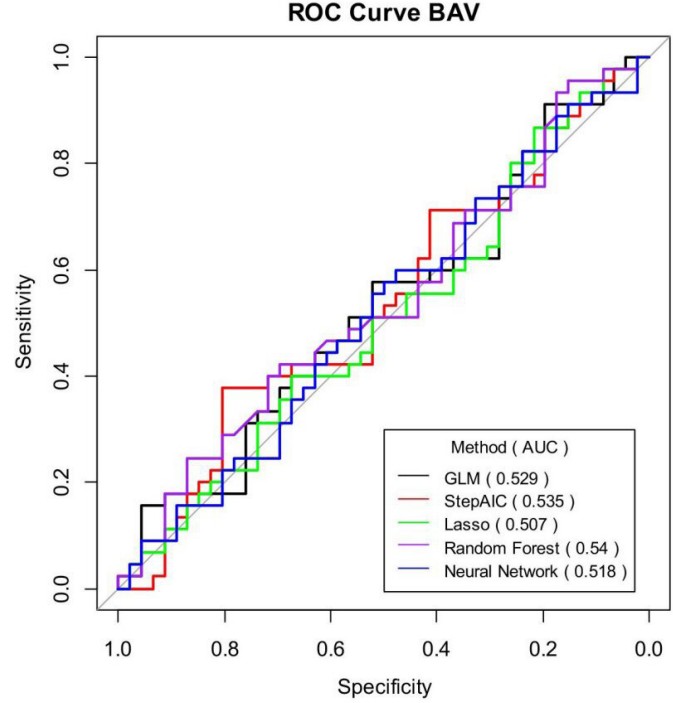

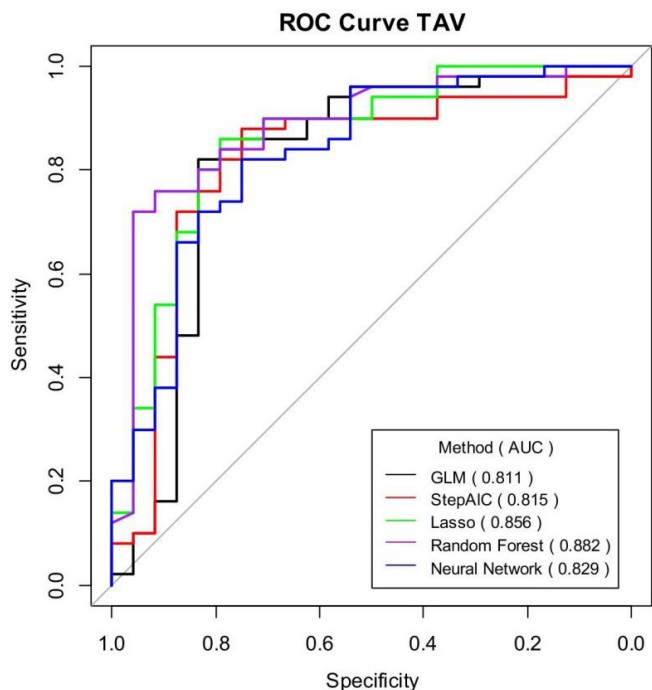

**Figure 2** The ROC (receiver operating characteristic) curve for risk prediction model across the predictive methods used. BAV, bicuspid aortic valve; TAV, tricuspid aortic valve.

of coronary artery disease and primarily not planned for another concomitant valve surgery, we report two key findings. First, the prediction of aortic dilatation using automated machine learning methods and traditional regression models on multidimensional clinical data was possible only among TAV individuals, and *not* in patients with BAV. This suggests that general clinical cardiovascular risk profiles play more important roles

during aortic dilation in TAV patients than in patients with a BAV, and further supports that aortopathy associated with BAV and TAV, respectively, is clearly distinct with different underlying aetiologies. Second, our study shows that the classification results were consistent for all machine learning algorithms and classic logistic regression models. This suggests that machine learning approaches might not outperform classical prediction methods in addressing complex interactions and non-linear relations between variables. This unexpected finding that classical statistics outperforms machine learning merits further exploration. This may be attributed to the distinct attributes and applications of each approach. Classical statistical methods, grounded in well-established mathematical theories, offer interpretability and inferential power, excelling in scenarios where assumptions are met and relationships between variables are linear or follow specific parametric forms. Conversely, machine learning techniques, leveraging algorithms to identify patterns in complex datasets, provide flexibility and the ability to capture non-linear relationships. They are adept at handling high-dimensional data, especially when the underlying data structure is intricate or not well defined.

The strongest predictor for ascending aortic dilation in TAV was the absence of AS. The other contributors to the prediction of aortic dilation in TAV are shown in online supplemental etable 6.

This is in accordance with our previous observations that surgical patients with AS and ascending aortic dilation almost exclusively have a BAV.[18] Whether this implies that biological processes associated with the development of AS in TAV also contribute to ascending aortic stability needs to be further elucidated. AS is commonly caused by progressive calcification of the aortic valve and increase in prevalence with age. In our cohort, TAV patients with dilated ascending aortas were significantly younger than TAV patients with non-dilated ascending aortas, which may be one contributing factor to the observed association and possibly reflects the surgical nature of the cohort. However, omitting patients with AS from the analysis still resulted in significant predictive values, although with worse discrimination, indicating that age alone cannot explain the association. Other strong contributors to the prediction model were diabetes and hsCRP. A negative association between diabetes and aneurysm in the abdominal aorta is well documented.[19] It has also been suggested that metformin prescription may associate with decreased risk of aortic dilation and that the molecular mechanism involves a metformin-induced reduction in aortic inflammation.[20 21] Similarly, an elevated hsCRP has been described in AAA patients and found to be an independent risk factor for AAA.[22 23] Moreover, we have previously shown an inflammatory gene expression profile in dilated aortic tissue from TAV but not BAV patients.[7] In the present study, there was a borderline significance of reduced hsCRP levels in BAV patients with dilated aorta. This together implies that aortic dilation in TAV, in

these aspects, may be more similar to aneurysm of the abdominal aorta than BAV aortic dilatation.

The lack of a good model for risk prediction of BAV-associated aortopathy raises the question, which other contributing factors may be of importance for aneurysm formation and development in these individuals? Two main hypotheses have been put forth in the literature.

First, an altered flow in the proximal part of the ascending aorta due to the valve malformation itself has been suggested to provoke aortic dilation. Also, different BAV morphotypes, that is, cusp fusion pattern, have been shown to cause flow disturbances that affect the aorta in morphotype-dependent ways.[24] In our study, flow characteristics were not included in the prediction model, which possibly could have influenced the results. However, we could not see any difference in the presence of aortic dilatation between different morphotypes, that is, true BAV, LN, RN or RL cusp fusion (table 1). Of note, other factors may also influence aortic flow patterns, for example, eccentricity of valve opening due to valve disease, or vessel stiffness. Indeed, it has been suggested that AS significantly alters aortic haemodynamic and wall shear stress, independent of aortic valve phenotype.

Second, the genetic contribution may override the influence of traditional risk factors to aneurysm development in patients with BAV. Although specific gene(s) and/or mutation(s) underlying BAV and BAV-associated aortopathy are still to be unravelled, several genes have been shown to be associated with both BAV formation and concomitant aortopathy in mice and humans (p2 of Ref 25).[26 27] Moreover, a high heritability of BAV and/or other cardiovascular malformations have been demonstrated using segregation patterns in families, with a heritability of BAV and BAV together with other cardiovascular malformations being as high as 89% and 75%, respectively.[28] A third, most likely, possibility is that both genetic factors and abnormal haemodynamic burden play central roles in BAV-associated aortopathy, interacting with each other and thereby contributing to aortic dilatation.

Further dissecting differences between patients with non-dilated and dilated ascending aortas we found that, among others, PP, AS, AR and diabetes were associated with dilatation in BAV. PP is a well-known risk factor for CVD and the clinical manifestation of increased vascular stiffness.[29] Surprisingly, PP was higher in BAV patients with a non-dilated aorta, which may seem counterintuitive since AR is associated with both increased PP and aortic dilatation. However, it may be speculated that the higher PP seen in these patients may rely on structural changes due to an increased haemodynamic burden associated with a BAV. In line with this, we have previously shown that BAV patients have a qualitative collagen defect in their ascending aorta, signified by a different collagen glycation compared with TAV patients and suggestive of an altered non-enzymatic collagen crosslinking.[30] Interestingly, we have also shown that dilated ascending aorta of BAV patients display an increased collagen-related

stiffness compared with TAV patients.[31] Furthermore, The Strong Heart Study could also show that in patients free of prevalent coronary heart disease, aortic root dilatation was, at a given diastolic blood pressure and stroke volume, associated with lower PP.[32] In our study, Angina pectoris appears to be more prevalent in individuals with non-dilated aortas. This observation can be elucidated by the exclusion criterion of significant coronary artery stenosis, as defined in our study. Despite this exclusion criterion, angina is reported as a symptomatic manifestation. It is noteworthy that angina may be symptomatic of AS or left ventricular hypertrophy induced by AS. The latter condition is more frequently encountered in isolation rather than in combination with proximal aortic dilatation. Additionally, our findings indicate a higher prevalence of angina, as well as other cardiovascular conditions (such as stroke, previous myocardial infarction, etc), and the use of medications in the group with TAVs, where the patients are older compared with the BAV group.

Additionally, accelerated vascular ageing and increased arterial stiffness have previously been described in patients with diabetes, the proportion of which was higher among BAV patients with non-dilated aortas.[33] However, we and others have previously demonstrated an increased vascular inflammation in dilated aorta in TAV but not BAV patients, suggesting that other mechanisms could be involved in the protective role of diabetes on aneurysm formation in BAV patients.[7 34] The association between dilatation and valve disease was not as pronounced in BAV as in TAV patients, although AS was also negatively associated with dilatation in BAV, as previously found.[35] It may be speculated that the presence of AS increases flow velocities and blood pressure in the ascending aorta, thereby stimulating vascular remodelling and strengthening of the aortic wall. Whether this hampers the process of dilatation remains to be answered. The relation between degree of stenosis and width of the ascending aorta is complex, and a previous study found mid-ascending dilatation proportional to valve gradient when patients with small aortas were excluded.[35]

Our findings raise the issue of how to identify and implement prevention of aortopathy in BAV patients in a clinical setting. So far, clinicians have focused on aortic valve function and aortic dimensions to indicate cardiac surgery and recommend annual follow-up in asymptomatic patients to screen for associated aortopathy.[36] High importance has been given to the morphology of the valve, although in our study, dilatation in BAV did not show any significant association with valve morphology. Of note, previous studies establishing an association between BAV cusp fusion and clinical outcomes relied on small sample size based on imaging diagnostic rather than anatomic diagnosis.[37]

## Study strengths and limitations

In this study, comprehensive clinical data, including blood sampling as well as epidemiological data, were used in the analysis of in total 1034 individuals (543 BAV and 491 TAV). The morphology of aortic valves was evaluated by visual inspection during open-heart surgery, which is a major strength compared with only echocardiography in terms of reliability.

A few limitations must however be highlighted. First, by design, only individuals devoid of significant coronary artery disease were included, which may introduce a selection bias and an overestimation of the prevalence of subjects with aneurysm. Second, as this is a population-based surgical cohort, it is possible that our study population included BAV and TAV patients with worse outcomes compared with their counterparts of similar age. This should however not affect the associations between valve type and aortic dilatation. Third, TAV patients with non-dilated aortas were significantly older than TAV individuals with a dilated aorta, which could possibly explain the higher degree of patients with AS in this group. Lastly, our study is a single-centre and monoethnicity study. Therefore, our results may not be generalisable to other population of BAV/TAV.

To conclude, using automated machine learning algorithms and classic logistic regression models, we demonstrated that in TAV patients, cardiovascular risk profiles appear to be more predictive of aortopathy than in BAV patients. The good performance of the TAV classifier also after exclusion of AS offers important implications for better targeting TAV individuals who are of a high risk of developing aneurysm. The lack of good models to develop clinical classifiers of BAV-associated aortopathy strengthens the focus of genetics and/or flow as important contributing factors to aneurysm development in these individuals.

**Author affiliations**
[1]Cardiovascular Medicine Unit, Center for Molecular Medicine, Department of Medicine, Karolinska Institutet, Stockholm, Sweden
[2]Hôpitaux universitaires de Genève, Genève, Switzerland
[3]Clinical Physiology Unit, Department of Molecular Medicine and Surgery, Karolinska Institutet, Stockholm, Sweden
[4]Cardiothoracic Surgery Unit, Department of Molecular Medicine and Surgery, Karolinska Institutet, Stockholm, Sweden

**Contributors** BG, AF-C, PE and HMB had full access to all the data in the study and took responsibility for the integrity of the data and the accuracy of the data analysis. Study concept and design: BG, PE and HMB. Acquisition, analysis or interpretation of data: all authors. Drafting of the manuscript: BG. Critical revision of the manuscript for important intellectual content: all authors. Statistical analysis: BG and MV. Study supervision: BG, PE and HMB.BG, PE and HMB accepts full responsibility for the conduct of the study, had access to the data, and controlled the decision to publish.

**Funding** The work was supported by the Swedish Research Council (12660); the Swedish Heart-Lung Foundation (20180451, 20170669); the Stockholm County Council (20180072); La Marató de TV3 (20151330); and a donation by Fredrik Lundberg.

**Competing interests** None declared.

**Patient and public involvement** Patients and/or the public were not involved in the design, or conduct, or reporting, or dissemination plans of this research.

**Patient consent for publication** Consent obtained directly from patient(s)

**Ethics approval** The study was approved by the Regional Ethics Review Board (application number 2006/784-31/1; 2012/1633-31/4), Stockholm, Sweden. Oral

and written informed consent was obtained from all patients according to the Declaration of Helsinki.

**Provenance and peer review**  Not commissioned; externally peer reviewed.

**Data availability statement**  Data are available upon reasonable request. The data of this article are available upon reasonable request to the corresponding author.

**ORCID iDs**
Bamba Gaye http://orcid.org/0000-0002-5516-4665
Christian Olsson http://orcid.org/0000-0002-5677-0747
Hanna M Björck http://orcid.org/0000-0002-9155-3609

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
