## [Reviewer comments · BMJ Open]

ARTICLE DETAILS

TITLE (PROVISIONAL)	Predictive Machine Learning Models for Ascending Aortic Dilatation in Patients with Bicuspid and Tricuspid Aortic Valves Undergoing Cardiothoracic Surgery: A prospective, single-center and observational study
AUTHORS	Gaye, Bamba; Vignac, Maxime; Gådin, Jesper R.; Ladouceur, Magalie; Caidahl, Kenneth; Olsson, Christian; Franco-Cereceda, Anders; Eriksson, Per; Björck, Hanna

VERSION 1 – REVIEW

REVIEWER	Kong, William National University Singapore
REVIEW RETURNED	25-Nov-2022

GENERAL COMMENTS	Gaye and co-workers had performed an analysis using machine learning algorithms and classic logistic regression models, and multiple variable selection methodologies to identify predictors of high risk of ascending aortic dilatation between the BAV and the TAV population. They found that Cardiovascular risk profiles appear to be more predictive of aortopathy in TAV but not in BAV. The paper is well written however i have a few comments: 1.) There was no mentioning of definition of hypertension, diabetes, hyperlipidemia and heart failure in the method column, and the author should referenced some papers for the above definition.2.) there was also no mentioning about the consent process of the study- whether waiver of consent or full consent was needed.3.) there were only three drugs mentioned in the tables but lacking of other anti hypertensive e.g- calcium channel blockers, ARBs and diuretics4.) the author should, if he is interested, to build another table or uni- and multi variable analysis of different segment of the aorta between the BAV and TAV groups.5.) to further improve the discussion, the author should improve another paragraph to discuss the sex differences of aortopathy between BAV and TAV, in reference to recent publications by a.) William K.F. Kong et al, Sex differences in bicuspid aortic valve disease, Progress in Cardiovascular Diseases, Volume 63, Issue 4, 2020, Pages 452-456, and b.) Michelena et al. Sex Differences in Bicuspid Aortic Valve Adults Who Deserves Our Attention, Men or Women? Circulation: Cardiovascular Imaging Volume 10, Issue 3, March 2017 https://doi.org/10.1161/CIRCIMAGING.117.0061236.) the author should mention another important limitation of the paper which the study is a single centre and mono- ethnicity study which the conclusion may not be not relevant to the general population of BAV/TAV.
--

REVIEWER	Hardikar, Ashutosh Royal Hobart Hospital, Cardiothoracic Surgery
REVIEW RETURNED	10-Dec-2022

GENERAL COMMENTS	1] 1034 patients selected from 2007 to 2017 seems a bit selective for a major cardiac surgical unit. Could we confirm if all patients with aortic valve disease and/or aortic pathology were chosen? 2] Similar databases of combined BAV and TAV tend to have a much larger proportion of TAV cases. Your database has 52% BAV cases. Is there a reason for the BAV preponderance if the database was to include all aortic valve cases? 3] 4. An ascending aorta of over 4 cm was called aortic dilatation. This measurement was taken at 4 cm from the level of aortic valve. In intrathoracic aneurysms, around 15-25% or aneurysms are restricted to aortic sinuses. By the definition applied in this model, they would be missed. How have the authors accounted for this? 4] BAV patients have higher BSA but lower BMI. The e-table 1 indicates that the BAV patients were taller. Does this have implications in the indexation according to height? 5] Angina pectoris seems to be commoner with non-dilated aortas. What could be the possible explanation or correlation? Would that link to protective effect of statins or anti-anginal medications? 6] 71% of TAV patients had dilated ascending aorta. If consecutive patients of aortic valve disease and / or aortic dilatation were included, this seems a very high percentage of patients being referred for surgery had aortic dilatation. Does that mean that in your institute majority of isolated aortic valve disease patients underwent TAVI rather than SAVR? What proportion of isolated AV disease at your hospital underwent TAVI and hence could have a higher proportion of Aortic regurgitation cases in your series?
---

REVIEWER	Raghupathi, Wullianallur Fordham University
REVIEW RETURNED	25-Feb-2023

GENERAL COMMENTS	Some narrative about comparison of statistical with ML - logistic etc. could be included. The statement that classic statistical outperforms ML is a bit dramatic. Additional scope & limitations discussion should be included.
--

VERSION 1 – AUTHOR RESPONSE

Reviewer #1:

Dr. William Kong, National University Singapore

Comments to the Author:

Gaye and co-workers had performed an analysis using machine learning algorithms and classic logistic regression models, and multiple variable selection methodologies to identify predictors of high risk of ascending aortic dilatation between the BAV and the TAV population.

They found that Cardiovascular risk profiles appear to be more predictive of aortopathy in TAV but not in BAV.

The paper is well written however i have a few comments:

Our answer: *We would like to thank the reviewer for the encouraging comments.*

Reviewer #1 Comment 1: There was no mentioning of definition of hypertension, diabetes, hyperlipidemia and heart failure in the method column, and the author should referenced some papers for the above definition.

Our Answer:

We thank the reviewer for raising this point. This is included in the revised version of the manuscript.

Reviewer #1 Comment 2: there was also no mentioning about the consent process of the study- whether waiver of consent or full consent was needed.

Our Answer:

*We thank the reviewer for raising this major ethical point. Information about the consent process of the study is available in the **Ethics Approval Statement** section: (see below)*

“The study was approved by the Regional Ethics Review Board (application number 2006/784-31/1; 2012/1633-31/4), Stockholm, Sweden. Oral and written informed consent were obtained from all patients according to the declaration of Helsinki.”

Reviewer #1 Comment 3: there were only three drugs mentioned in the tables but lacking of other anti-hypertensive e.g- calcium channel blockers, ARBs and diuretics

Our Answer:

*We thank the reviewer for raising this point. Information on other anti-hypertensive drugs is presented in the table below. We added this table in the supplemental material and revised the manuscript accordingly. **See the Caption of Table 1.***

“Caption: Chi square test was used for categorical variables; ANOVA for normal-distributed continuous variables; Kruskal-Wallis test for non-normal continuous variables. BSA, body surface area; BMI, body mass index; hsCRP, high sensitive C-reactive protein; LDL, low density lipoprotein; STJ, sinotubular junction; AAA, abdominal aortic aneurysm. ***Information on other anti-hypertensive drugs is presented in e- table 5.”**

Reviewer #1 Comment 4: the author should, if he is interested, to build another table or uni- and multi variable analysis of different segment of the aorta between the BAV and TAV groups.

Our Answer:

*We thank the reviewer for raising this point. We agree that it would be interesting to even explore how the **Clinical classifiers could help identifying dilatation of different segment of the aorta in patients with bicuspid versus tricuspid aortic valve.** However, even if they share some risk factors,*

the pathophysiology of aneurysm is not the same in the different segments of the aorta. (ref) We believe that ascending aortic dilatation should remain the aim of the manuscript.

Reviewer #1 Comment 5: to further improve the discussion, the author should improve another paragraph to discuss the sex differences of aortopathy between BAV and TAV, in reference to recent publications by a.) William K.F. Kong et al, Sex differences in bicuspid aortic valve disease, Progress in Cardiovascular Diseases, Volume 63, Issue 4, 2020, Pages 452-456, and b.) Michelena et al. Sex Differences in Bicuspid Aortic Valve Adults Who Deserves Our Attention, Men or Women? Circulation: Cardiovascular Imaging Volume 10, Issue 3, March 2017 <https://doi.org/10.1161/CIRCIMAGING.117.006123>

Our Answer:

We thank the reviewer for raising this very relevant point. Sex differences in aortopathy and valve disease between BAV and TAV have already been studied in the same cohort (Vignac et al. The Annals of Thoracic Surgery. 2022) . In this manuscript, we discussed both the manuscript of William K.F. Kong et al. and Michelena et al.)

Reviewer #1 Comment 6: the author should mention another important limitation of the paper which the study is a single centre and mono- ethnicity study which the conclusion may not be not relevant to the general population of BAV/TAV.

Our Answer:

We thank the reviewer for raising this major point. We emphasised this point in the proposed reviewed version. See below:

*“A few limitations must however be highlighted. Firstly, by design, only individuals devoid of significant coronary artery disease were included, which may introduce a selection bias and an overestimation of the prevalence of subjects with aneurysm. Secondly, as this is a population based surgical cohort, it is possible that our study population included BAV and TAV patients with worse outcomes compared to their counterparts of similar age. This should however not affect the associations between valve type and aortic dilatation. Thirdly, TAV patients with non-dilated aortas were significantly older than TAV individuals with a dilated aorta, which could possibly explain the higher degree of patients with aortic stenosis in this group. **Lastly our study is a single centre and mono- ethnicity study. Therefore, our results may not be generalizable to other population of BAV/TAV.**”*

Reviewer #2:

Dr. Ashutosh Hardikar, Royal Hobart Hospital
Comments to the Author:

Reviewer #2 Comment 1: 1034 patients selected from 2007 to 2017 seems a bit selective for a major cardiac surgical unit. Could we confirm if all patients with aortic valve disease and/or aortic pathology were chosen?

Our Answer:

We thank the reviewer for raising this point. Unfortunately, it will be very difficult to answer the question “how many patients with ascending aneurysm OR AS OR AR OR any combination thereof was operated during the 11-year study period? Simple ascending aneurysm is around 100/year, but isolated aortic valves are probably also around 250/year and then we would need to tease out for example endocarditis, valve coronary artery bypass graft, acute cases and so on. The term “consecutive patients” may be misleading.

Over all, the number of patients is around 100/year. Maybe we need to clarify the total number of eligible patients (i.e., any aortic valve and/or proximal aortic procedure)? An increased proportion of TAV patients treated by TAVI is very probable, resulting in some overrepresentation of BAV in our surgical cohort. Most probably, other selection processes are also at play.

Also, normal root with dilated ascending distal to the STJ is quite common. Yet, most commonly I think both coexist, representing a continuum from the root towards the arch.

Reviewer #2 Comment 2: Similar databases of combined BAV and TAV tend to have a much larger proportion of TAV cases. Your database has 52% BAV cases. Is there a reason for the BAV preponderance if the database was to include all aortic valve cases?

Our Answer:

We thank the reviewer for raising this point. In fact, there is no “database”, there are the ASAP and DAVACAA study patient cohorts (described in the methodology) and they were based on the inclusion and exclusion criteria we refer to. By applying significant coronary artery stenosis as an exclusion criterion, elderly patients, in which the combination is more prevalent, were more often excluded, as reflected as comparatively low mean age (60 and 70 years for BAV and TAV, respectively), in turn favoring the inclusion of more BAV patients.

Reviewer #2 Comment 3: An ascending aorta of over 4 cm was called aortic dilatation. This measurement was taken at 4 cm from the level of aortic valve. In intrathoracic aneurysms, around 15-25% or aneurysms are restricted to aortic sinuses. By the definition applied in this model, they would be missed. How have the authors accounted for this?

Our Answer:

We thank the reviewer for raising this point. Patients with root dilatation are in fact included. Maybe we should just apologize and rephrase to “proximal aortic dilatation” instead of “ascending”, to allow such patients to concur with the inclusion criteria.

Reviewer #2 Comment 4: BAV patients have higher BSA but lower BMI. The e-table 1 indicates that the BAV patients were taller. Does this have implications in the indexation according to height?

Our Answer:

We thank the reviewer for raising this point. Higher BSA despite lower BMI in BAV patients may be explained by statistically height differences. In our models, we used statistical methods such as clustering and Principal Component Analysis to determine the most relevant variables to include in the analysis (Supplemental e-Figure 2). Height appeared to be more relevant for our models.

Reviewer #2 Comment 5: Angina pectoris seems to be commoner with non-dilated aortas. What could be the possible explanation or correlation? Would that link to protective effect of statins or anti-anginal medications?

Our Answer:

By definition, significant coronary artery stenosis was an exclusion criterion. Nevertheless, angina is reported as a symptom. Angina could be a symptom from aortic stenosis (or the left ventricular hypertrophy caused by AS), which in turn is more common as an isolated condition rather than a combination with proximal aortic dilatation. Furthermore, angina as well as both other cardiovascular conditions (stroke, previous MI and so on) as medications were all more common in the group (TAV) with the oldest patients (mean 70 vs mean 60 for BAV).

Reviewer #2 Comment 6: 71% of TAV patients had dilated ascending aorta. If consecutive patients of aortic valve disease and / or aortic dilatation were included, this seems a very high percentage of patients being referred for surgery had aortic dilatation. Does that mean that in your institute majority of isolated aortic valve disease patients underwent TAVI rather than SAVR? What proportion of isolated AV disease at your hospital underwent TAVI and hence could have a higher proportion of Aortic regurgitation cases in your series?

Our Answer:

We thank the reviewer for raising this point. Dilated ascending aorta was defined as aortic diameter above 40mm. This sensitive cut off leads to higher prevalence of aneurysm.

Reviewer #3:

Dr. Wullianallur Raghupathi, Fordham University

Comments to the Author:

Reviewer #3 Comment 1: Some narrative about comparison of statistical with ML - logistic etc. could be included.

Our Answer:

We thank the reviewer for raising this point. This point was addressed in the revised manuscript.

Reviewer #3 Comment 2: The statement that classic statistical outperforms ML is a bit dramatic. Additional scope & limitations discussion should be included.

Our Answer:

We thank the reviewer for raising this point. This point was addressed in the revised manuscript.